# Perceived barriers to timely treatment initiation and social support status among women with breast cancer in Ethiopia

Bethel Teshome[1]◉, Josephin Trabitzsch[2]◉, Tsion Afework◉[1], Adamu Addissie[1,2], Mirgissa Kaba[1], Eva Johanna Kantelhardt◉[2,3]*, Sefonias Getachew[1,2]

1 Department of Preventive Medicine, School of Public Health, Addis Ababa University, Addis Ababa, Ethiopia, 2 Institute for Medical Epidemiology, Biometrics and Informatics, Martin-Luther-University Halle-Wittenberg, Halle Saale, Germany, 3 Department of Gynaecology, Martin-Luther-University Halle-Wittenberg, Halle Saale, Germany

◉ These authors contributed equally to this work.
* eva.kantelhardt@uk-halle.de

**Data Availability Statement:** All relevant data are within the manuscript and its Supporting Information files.

## Abstract

Timely care is essential to increase breast cancer survival. However, patients in Ethiopia still face multilevel barriers on their pathway to timely treatment initiation. This cross-sectional study at Tikur Anbessa Specialized Hospital Oncology Unit in Addis Ababa assessed systemic treatment initiation intervals of breast cancer patients and quantified the impact of socio-demographic and clinical factors, perceived barriers, and the patients´ perceived social support status on timely systemic treatment initiation (chemotherapy or hormonal therapy). A structured questionnaire was designed based on Pechansky´s "Concept of Access". Applying simple and multivariate logistic regression we analysed the influence of patients´ characteristics as well as their perceived barriers on timely treatment initiation. We measured social support with the Multidimensional Score of Perceived Social Support (MSPSS) and used the Wilcoxon Rank-Sum Test to assess its relationship with timely treatment initiation. Of 196 patients included into the study, 53% received systemic treatment within 90 days of their pathological diagnosis–the median treatment initiation interval was 85 days (IQR 123.5). Older women and patients diagnosed at late stages had higher odds of timely treatment initiation. Not being able to pay for services and lack of transport were most often perceived as barriers towards timely care. However, none of the perceived barriers showed a substantial influence on timely treatment initiation in the multivariate regression model. The patients´ perceived social support was found to be high, with an average MSPSS score of 73 out of 84 (SD 13,63). No impact of the perceived social support status on timely treatment initiation was found. The percentage of breast cancer patients waiting longer than 90 days from pathological diagnosis to systemic treatment initiation in Ethiopia remains unacceptably high. While women generally feel well supported by their social environment, costs and accessibility of treatment are perceived to be major barriers towards timely treatment initiation.

**Funding:** This study was funded within the Masters Program of the School of Public Health, Addis Ababa University. BT received the fund from the University. The funding body was not involved in the design of the study and collection, analysis and interpretation of data and in writing the manuscript.

**Competing interests:** The authors have declared that no competing interests exist.

## Introduction

Breast cancer is the most frequent cancer in women in Sub-Saharan Africa (SSA) [1] and is becoming a major health burden, as life expectancy, reproductive patterns and lifestyles are changing [2]. In Ethiopia, breast cancer is currently estimated to make up for 33% of all new cancer cases in women and 17% of all cancer deaths [1].

Survival rates differ significantly between low- or middle-income and high-income countries [3]. This has been largely attributed to late stages at diagnosis and extended intervals until treatment initiation in countries with high cancer mortality [3].

Social support is one factor which has been particularly emphasised to influence the women´s pathways throughout the disease. In the US, studies found a two-fold increased risk of mortality from breast cancer for women with low levels of social integration and lower odds of treatment initiation for socially isolated women [4, 5].

Research on cancer patient pathways in low- and middle-income countries has been focusing on the intervals between first symptom appraisal and diagnosis [6]. Even though time between diagnosis and treatment initiation has been equally established to influence the outcome of breast cancer therapy [7], there is a paucity of studies reporting on reasons for delay in treatment initiation in low- and middle income countries [6, 8]. Additionally, the patients´ own perception of barriers as well as the role of social support within the cancer care continuum have been mainly addressed on an exploratory level [9–11].

In this study we determined systemic treatment initiation intervals of breast cancer patients at a tertiary referral hospital in Ethiopia and assessed the impact of socio-demographic and clinical factors, as well as patients´ perceived barriers and their perceived social support status on timely treatment initiation.

## Methods

### Study design and setting

This cross-sectional study was conducted between March and May 2018 at the Oncology Unit of Tikur Anbessa Specialized Hospital in Addis Ababa, Ethiopia. The hospital serves as a third-level governmental referral hospital and is the only hospital in the country offering fully comprehensive cancer treatment. Applying the "Model of Pathways to Treatment", we defined the treatment initiation interval as the period between pathological diagnosis and the begin of systemic treatment (chemotherapy or hormonal therapy) [12].

The primary outcome of the study was timely systemic treatment initiation, which we defined as ≤ 90 days between pathological diagnosis and systemic treatment initiation. In the absence of official guidelines, this cut-off value was chosen based on similar studies in comparable settings [13, 14]. Patients receiving radiotherapy were not included into the study, as, with only one radiotherapy machine in the country, the number of breast cancer patients being treated with radiation within the study period was judged to be too small to receive reliable results.

### Population, sample, and data collection

The study constituted pathologically diagnosed breast cancer patients who were on systemic treatment or follow-up care and 18 years or older at time of data collection. Patients who were critically sick and unable to communicate were excluded. To compensate for large outliers, we also excluded patients whose treatment initiation intervals could not be determined or whose treatment initiation interval was longer than two years.

Simple random sampling was used to identify study participants; in advance of each day of data collection 70% of all patients appointed for treatment or follow-up were randomly

selected to participate in the study. Women were interviewed face-to-face by trained oncology nurses before their appointments at the oncology unit. A structured questionnaire was prepared in cooperation with a senior oncologist in easily understandable local language (Amharic), pretested by the principal investigator on 10% of the original sample size (n = 32) and adapted accordingly to assure understandability (S1 Questionnaire). Questions addressing perceived barriers to timely treatment initiation were designed according to the "Concept of Access" by Pechansky and Thomas. [15]. In this model access is defined to summarize five specific "dimensions of fit" between patients and the health system: Availability, accessibility, accommodation, affordability, and acceptability. The first four dimensions can be summarised by Walter´s *healthcare providers and system factors*, whereas *acceptability* mainly describes *patient factors*, such as social environment, culture, and previous experiences. Clinical data generated through interviews were triangulated with data from patient charts where possible.

The 12-item Multidimensional Scale of Perceived Social Support (MSPSS) was used to measure the perceived social support of patients. The tool was designed and validated to evaluate the adequacy of social support received from family, friends and significant other persons [16]. Four questions each cover one of the three items and are later summed up to build the score. A seven-point Likert scale enables patients to choose between 1 (= very strongly disagree) and 7 (= very strongly agree).

## Data analysis

Data was entered into EpiData and transferred to R Version 4.0.3 for analysis. Descriptive analysis was applied to calculate treatment initiation intervals, the patients´ perception of barriers to timely treatment as well as social support. Because of large outliers, median and interquartile range (IQR) were used to describe intervals. For patients, whose method and date of diagnosis were unknown, we assumed date of surgery to be date of pathological diagnosis.

Socio-demographic and medical factors influencing timely treatment initiation were modelled using simple and multivariate logistic regression. A separate model was applied to assess the influence of perceived barriers, which was adjusted for age and stage. Crude odds ratios (COR) and adjusted odds ratios (AOR) are presented with 95% confidence intervals.

The Wilcoxon rank-sum test was performed to analyse the relationship between the treatment initiation interval and social support. This non-parametric test was chosen, as the assumption of normality was not fulfilled.

## Ethical considerations

Ethical clearance was obtained from the Ethical Review Committee at the School of Public Health of Addis Ababa University. Written informed consent was obtained from all participants in advance of the interview.

## Results

From 302 patients, 106 patients were excluded from analysis due to missing dates of diagnosis or treatment. Of the 196 women included into the study, most were diagnosed at stage II or III (Table 1).

Use of alternative medicine (traditional or spiritual healers) between diagnosis and systemic treatment initiation was reported by 34% of all patients.

The median treatment initiation interval in the study cohort was 85 days (IQR 123.5) (Table 2).

Based on the cut-off value of 90 days, 53% of the patients received systemic treatment in time, while 47% received treatment later than 90 days after their diagnosis.

**Table 1. Socio-demographic and clinical characteristics of study cohort.**

| Variable | Frequency | Percentage (%) |
|---|---|---|
| **Number of patients** | 196 | 100 |
| **Age[a]** (in years) | | |
| ≤ 45 | 114 | 58.2 |
| > 45 | 58 | 29.6 |
| Unknown | 24 | 12.2 |
| **Residence** | | |
| In town | 106 | 54.1 |
| Out of town | 84 | 42.9 |
| Unknown | 6 | 3.1 |
| **Marital status** | | |
| Married | 113 | 57.7 |
| Not married | 83 | 42.3 |
| **Education** | | |
| Primary school or lower | 58 | 29.6 |
| Secondary school or higher | 88 | 44.9 |
| Unknown | 50 | 25.5 |
| **Stage** | | |
| I | 29 | 14.8 |
| II | 64 | 32.7 |
| III | 67 | 34.2 |
| IV | 28 | 14.3 |
| Unknown | 8 | 4.1 |
| **Use of alternative treatment[b]** | | |
| Yes | 66 | 33.7 |
| No | 129 | 65.8 |
| Unknown | 1 | 0.5 |

[a] 45 years cut-off was chosen as this is commonly judged the median age of menopause in this population.

[b] Alternative treatment includes visits at traditional or spiritual healers.

In the regression models, women older than 45 years had higher odds of timely treatment initiation (COR 1.47 and AOR 3.18) than younger patients (Table 3).

Stage was also associated with timely treatment initiation–in the multivariate model women diagnosed with stages II, III or IV had 3.5 to 4.5 times the odds to initiate treatment in time compared with patients diagnosed at stage I.

Patients´ perceived barriers towards timely treatment initiation were grouped into four items based on the *Concept of Access to Care* (Table 4).

*Affordability* and *accessibility* were perceived as most important barriers: 66% of all patients considered not being able to pay for the service had been a barrier towards timely care, and 54% perceived lack of transport as a barrier. Concerning *accommodation and acceptability*, long waiting times were perceived by 48% of all women as an important barrier, while 38% reported to have been hindered by their fear of the treatment´s side effects.

In the multivariate regression, patients who had not perceived lack of transport and long waiting times as barriers had higher odds of timely treatment initiation (AOR 2.08 and 1.31)– however those findings were not significant on a 5% significance level (p-values 0.09 and 0.54).

The patients´ scores in the Multidimensional Score of Perceived Social Support were found to be high, with an average total score of 73 out of 84 (SD 13,63) (Table 5).

**Table 2. Systemic treatment initiation interval by treatment and type of administration.**

| | n (%) | Median (IQR) | Range |
|---|---|---|---|
| **All patients** | 196 (100) | 85 (123.5) | 1–726 |
| Timely | 104 (53.1) | 32.5 (43.2) | 1–90 |
| Not timely | 92 (46.9) | 158.5 (170) | 91–726 |
| **Type of treatment** | | | |
| Chemotherapy | 169 (86.2) | 85 (112) | 1–697 |
| Hormonal therapy | 21 (10.7) | 85 (310) | 1–726 |
| Unknown [a] | 6 (3.1) | 65.5 (73.2) | 12–110 |
| **Type of treatment administration** | | | |
| Adjuvant | 145 (74) | 85 (108) | 1–697 |
| Neoadjuvant | 35 (17.9) | 80 (159.5) | 1–726 |
| Without surgery[b] | 9 (4.6) | 32 (112) | 8–184 |
| Unknown | 7 (3.6) | 160 (420.5) | 8–539 |

*IQR* interquartile range

[a] Unknown due to contradicting information between patients´ charts and questionnaire data.

[b] Palliative intent.

**Table 3. Simple and multivariate logistic regression for factors associated with timely systemic treatment initiation ($\leq$ 90 days since pathological diagnosis) of breast cancer patients.**

| Characteristic | All | Timely (%) | COR (CI) | p[a] | AOR (CI) | p[b] |
|---|---|---|---|---|---|---|
| **Age (years)** | | | | | | |
| $\leq$45 | 114 | 58 (50.9) | Reference | | | |
| >45 | 58 | 35 (60.3) | 1.47 (0.77–2.79) | 0.24 | 3.18 (1.2–8.38) | 0.02 |
| **Stage** | | | | | | |
| I | 29 | 8 (27.6) | Reference | | | |
| II | 64 | 37 (57.8) | 3.6 (1.39–9.33) | 0.01 | 4.5 (1.29–15.72) | 0.02 |
| III | 67 | 39 (58.2) | 3.66 (1.42–9.44) | 0.01 | 3.62 (1.03–12.77) | 0.05 |
| IV | 28 | 16 (57.1) | 3.5 (1.16–10.58) | 0.03 | 3.52 (0.79–15.68) | 0.1 |
| **Residence** | | | | | | |
| Out of town | 84 | 41 (48.8) | Reference | | | |
| In town | 106 | 60 (56.6) | 1.37 (0.77–2.43) | 0.29 | 1.07 (0.48–2.43) | 0.86 |
| **Marital status** | | | | | | |
| Not married | 83 | 40 (48.2) | Reference | | | |
| Married | 113 | 64 (56.6) | 1.4 (0.79–2.48) | 0.24 | 1.08 (0.49–2.4) | 0.85 |
| **Education level** | | | | | | |
| Secondary or higher | 88 | 47 (53.4) | Reference | | | |
| Primary or lower | 58 | 35 (60.3) | 1.33 (0.68–2.6) | 0.41 | 1.1 (0.48–2.49) | 0.82 |
| **Use of alternative treatment[c]** | | | | | | |
| Yes | 129 | 65 (50.4) | Reference | | | |
| No | 66 | 38 (57.6) | 1.34 (0.73–2.43) | 0.34 | 1.59 (0.7–3.59) | 0.26 |

*COR* crude odds ratio, *CI* confidence interval, *AOR* adjusted odds ratio, *p* p-value

[a] P-value for simple regression models.

[b] P-value for multivariate regression model.

[c] Alternative treatment includes visits at traditional or spiritual healers.

**Table 4. Simple and multivariate regression modell of the influence of patients´ perceived barriers on timely systemic treatment initiation for breast cancer patients.**

| | Total (%[a]) | Timely (%[b]) | COR (CI) | p[c] | AOR[d] (CI) | p[e] |
|---|---|---|---|---|---|---|
| **AFFORDABILITY** | | | | | | |
| **Lack of money** | | | | | | |
| Important | 130 (66.3) | 63 (48.5) | Reference | | | |
| Not important | 64 (32.6) | 39 (60.9) | 1.66 (0.9–3.05) | 0.1 | 1.12 (0.47–2.67) | 0.79 |
| **ACCESSIBILITY** | | | | | | |
| **Lack of transport** | | | | | | |
| Important | 106 (54.1) | 51 (48.1) | Reference | | | |
| Not important | 86 (43.9) | 52 (60.5) | 1.65 (0.93–2.93) | 0.09 | 2.08 (0.88–4.91) | 0.09 |
| **ACCOMMODATION** | | | | | | |
| **Long waiting times** | | | | | | |
| Important | 94 (48.0) | 43 (45.7) | Reference | | | |
| Not important | 99 (50.5) | 60 (60.6) | 1.82 (1.03–3.23) | 0.04 | 1.31 (0.56–3.05) | 0.54 |
| **Nobody to look after children** | | | | | | |
| Important | 50 (25.5) | 26 (52) | Reference | | | |
| Not important | 141 (71.9) | 75 (53.2) | 1.05 (0.55–2) | 0.88 | 0.81 (0.29–2.24) | 0.68 |
| **Lack of time** | | | | | | |
| Important | 38 (19.4) | 22 (57.9) | Reference | | Reference | |
| Not important | 155 (79.1) | 81 (52.3) | 0.8 (0.39–1.63) | 0.53 | 0.38 (0.11–1.24) | 0.11 |
| **ACCEPTABILITY** | | | | | | |
| **Fear of side effects** | | | | | | |
| Important | 74 (37.8) | 36 (48.6) | Reference | | | |
| Not important | 120 (61.2) | 67 (55.8) | 1.33 (0.75–2.39) | 0.33 | 1.02 (0.44–2.39) | 0.96 |
| **Wanted to handle it by oneself** | | | | | | |
| Important | 59 (30.1) | 34 (57.6) | Reference | | | |
| Not important | 132 (67.3) | 66 (50) | 0.74 (0.4–1.37) | 0.33 | 0.9 (0.36–2.24) | 0.82 |
| **Embarrassment** | | | | | | |
| Important | 57 (29.1) | 25 (43.9) | Reference | | | |
| Not important | 138 (70.4) | 78 (56.5) | 1.66 (0.89–3.1) | 0.11 | 2.04 (0.84–4.98) | 0.12 |
| **Hope for disease to disappear by itself** | | | | | | |
| Important | 56 (28.6) | 28 (50) | Reference | | | |
| Not important | 135 (68.9) | 71 (52.6) | 1.11 (0.59–2.07) | 0.74 | 1.07 (0.41–2.79) | 0.9 |
| **Bad experiences with past treatment** | | | | | | |
| Important | 38 (19.4) | 20 (52.6) | Reference | | | |
| Not important | 155 (79.1) | 82 (52.9) | 1.01 (0.5–2.06) | 0.98 | 0.85 (0.29–2.55) | 0.77 |

*COR* crude odds ratio, *CI* confidence interval, *AOR* adjusted odds ratio, *p* p-value

[a] Percentages in relation to total number of patients (n = 196). Missing answers excluded.

[b] Row wise percentages.

[c] P-value for simple regression models.

[d] Additionally adjusted for age and stage.

[e] P-value for multivariate regression model.

Average scoring for support by the family and a significant other person was 25 of 28, and social support by friends slightly lower with a mean score of 22. There was no substantial difference in average scores between patients with timely treatment initiation and those with a treatment initiation interval longer than 90 days.

**Table 5. Multidimensional Score of Perceived Social Support (MSPSS) of breast cancer patients.**

|  | Family | Friends | Significant other | Total |
|---|---|---|---|---|
| **MSPSS max. score** | 28 | 28 | 28 | 84 |
| **Mean (SD)** | 25 (4.42) | 22 (6.37) | 25 (5.59) | 73 (13.63) |
| **Median (IQR)** | 27 (4) | 25 (9) | 27 (4) | 76 (13) |
| **p-value[a]** | 0.35 | 0.64 | 0.35 | 0.51 |

[a] Wilcoxon rank-sum test

## Discussion

In this retrospective Ethiopian breast cancer cohort, we found almost half of all patients to receive systemic treatment later than 90 days after their pathological diagnosis and quantified the importance of affordability and accessibility of systemic treatment in the perception of patients. Women with breast cancer at Tikur Anbessa Specialized Hospital Oncology Unit felt highly supported by their social environment, however we could not find an association between social support and timely treatment initiation. As common in countries in SSA, patients were predominantly diagnosed at young age and late cancer stages [3]. However, with 47.5% of the patients being diagnosed at stage I and II, a shift towards earlier stages at diagnosis was visible in comparison with a study from the same hospital in 2011 [17]. As there is still no regular screening for breast cancer available in Ethiopia, this dynamic might point towards higher awareness of breast cancer and improved access to diagnosis and care today compared with 2011. Rates of patients receiving surgery as well as the systemic treatment patterns observed in this study were in coherence with previous data from Addis Ababa [17].

In its "Guide to Cancer Early Diagnosis" the World Health Organization calls for a maximum of 90 days between symptom onset and treatment initiation [18]. The median of 85 days between diagnosis and systemic treatment initiation in our study (which does not include time from symptom onset until pathological diagnosis) shows that the WHO´s standard does not mirror reality for most breast cancer patients in Ethiopia. This observation has been reported by multiple studies from SSA [19]. A study based on the Addis Ababa Cancer Registry analysed treatment initiation intervals for all cancer entities and found a median time to therapy of 2.1 months [20]. However, this interval was measured from date of therapy planning until treatment initiation, which does not include time between diagnosis and planning of systemic therapy. In a comparable referral hospital in Botswana a median treatment initiation interval of 91 days was found [14].

The influence of age on treatment initiation has been discussed controversially in literature [8, 14]. A multi-centre prospective study on treatment initiation of breast cancer patients in Namibia, Nigeria and Uganda found women below 40 years having the lowest odds of initiating treatment within one year of diagnosis [21]. This observation supports the results from our study and might be explained by a less stable financial situation of younger women as well as larger responsibilities at home, with children and at work. However, our study did not collect adequate data to support or contradict this explanation.

Our observation that diagnosis at higher stages might be positively associated with timely treatment initiation is in line with findings from a cross-sectional study on pathways of breast cancer patients in South-Africa [11]. Patients with small tumours might be less aware of the seriousness of the disease; however, as breast cancer patients treated at early stages have considerably better outcomes than when treated in later stages [3], this finding is disconcerting. The inverse correlation of early stage with timely treatment initiation needs to be further

investigated and awareness must be raised among medical personnel to assure timeliness for these patients. Currently, there is no triage system in place for cancer patients.

Most patients perceived affordability and accessibility of care as most important barriers. While we could not find a correlation between perceived barriers and timely treatment initiation, the findings show the strong influence of the socio-economic status on timely treatment initiation in countries where treatment-costs are being paid out-of-pocket [21]. Even though patients without financial resources can get free treatment in government hospitals in Ethiopia, the bureaucratic hurdles to receive the so-called "poor papers" (proofing eligibility for free treatment), as well as the indirect costs linked to cancer treatment, still seem to hinder patients when aiming to receive treatment.

The steep increase in patient volumes at Tikur Anbessa Specialized Hospital over the last years [22] might be another reason why we found nearly half of all patients receiving systemic treatment more than three months after diagnosis. Long waiting times were considered a barrier towards timely treatment initiation by 48% of all patients, a finding which reinforces the government´s efforts to increase workforce and technical capacities by establishing multiple peripheral cancer centres within its Health Sector Transformation Plan.

Interestingly, factors concerning *acceptability* were generally perceived as less of a barrier. Although, every third women reported having tried alternative treatment in the course of her disease, patients' sentiments towards conventional medicine seemed generally positive.

The high level of social support patients reported in this study is consistent with another study from Ethiopia [23]. Women with breast cancer in Ethiopia have been found to have good social networks [24], which are possibly cushioning the negative impact of the disease and its treatment. This consistent high support might be an explanation, why our study could not find any correlation between the patient's social support status and their treatment initiation intervals. Notably, studies observing a quantifiable link between social support and delayed treatment initiation are exclusively from high-income-countries, such as the United States [4, 5], where health-system mediated barriers towards timely treatment are comparably smaller and social networks possibly weaker.

## Strengths and limitations

The question how to improve access to cancer treatment in resource-limited countries is complex and much debated in public health sciences. We consider it a strength of this study to have captured different aspects of this challenge within one breast cancer study cohort.

However, we had to exclude almost one-third of all patients from analysis due to missing data. This means the sample may not fully reflect the variety of patients´ situations. As the study is hospital-based and retrospective, it also allows for some bias of selection as well as recollection.

In addition, our data does not allow us to distinguish between a "patient interval" (delay in treatment due to the patient not making an appointment) and a "health care provider interval" (delay in treatment due to waiting times). This limits the study´s ability to draw conclusions on reasons for the delay in treatment initiation.

## Conclusion

In the last decade, policy makers, non-governmental organisations, and health researchers in Ethiopia have increased their attention on cancer, amounting in the publication of the first National Cancer Control Plan in 2015. The urgency to expand cancer care capacities is underlined by our finding that nearly half of all breast cancer patients received systemic treatment later than three months after diagnosis. Reasons why young patients and early stage tumors

are associated with delayed systemic treatment initiation clearly need further assessment to support these important patient groups.

Affordability and access to systemic cancer treatment were perceived as major barriers towards timely treatment initiation despite not being associated with actual delay. This reveals a dilemma faced by many healthcare systems in low- and middle-income countries that have to distribute financial resources between competing priorities. Since systemic cancer therapy is costly, international initiatives such as the World Health Organisation suggest joining forces to reduce the financial burden for these countries. The high-perceived level of social support underlines a great strength of the Ethiopian society which cannot be highlighted enough and sets an example for others.

## Supporting information

**S1 Dataset.**
(XLSX)

**S1 Questionnaire.**
(PDF)

## Author Contributions

**Conceptualization:** Bethel Teshome, Mirgissa Kaba, Sefonias Getachew.

**Data curation:** Josephin Trabitzsch, Tsion Afework.

**Formal analysis:** Josephin Trabitzsch.

**Funding acquisition:** Bethel Teshome.

**Investigation:** Bethel Teshome.

**Methodology:** Bethel Teshome, Josephin Trabitzsch, Sefonias Getachew.

**Supervision:** Adamu Addissie, Mirgissa Kaba, Eva Johanna Kantelhardt, Sefonias Getachew.

**Validation:** Eva Johanna Kantelhardt.

**Writing – original draft:** Josephin Trabitzsch.

**Writing – review & editing:** Bethel Teshome, Josephin Trabitzsch, Tsion Afework, Adamu Addissie, Mirgissa Kaba, Eva Johanna Kantelhardt, Sefonias Getachew.

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
