## [Decision Letter · Decision Letter 0]

9 Jul 2021

PONE-D-21-16175

Perceived barriers to timely treatment initiation and social support status among women with breast cancer in Ethiopia

PLOS ONE

Dear Dr. Eva Johanna Kantelhardt,

Thank you for submitting your manuscript to PLOS ONE. After careful consideration, we feel that it has merit but does not fully meet PLOS ONE’s publication criteria as it currently stands. Therefore, we invite you to submit a revised version of the manuscript that addresses the points raised during the review process.

We look forward to receiving your revised manuscript.

Kind regards,

Justyna Dominika Kowalska

Academic Editor

PLOS ONE

Journal Requirements:

2. Please include additional information regarding the survey or questionnaire used in the study and ensure that you have provided sufficient details that others could replicate the analyses. For instance, if you developed a questionnaire as part of this study and it is not under a copyright more restrictive than CC-BY, please include a copy, in both the original language and English, as Supporting Information. Moreover, please give more information on the development of the questionnaire, and explain how it was pre-tested.

Reviewers' comments:

Reviewer's Responses to Questions

**Comments to the Author**

1. Is the manuscript technically sound, and do the data support the conclusions?

Reviewer #1: Partly

Reviewer #2: Yes

2. Has the statistical analysis been performed appropriately and rigorously? 

Reviewer #1: No

Reviewer #2: Yes

3. Have the authors made all data underlying the findings in their manuscript fully available?

Reviewer #1: Yes

Reviewer #2: Yes

4. Is the manuscript presented in an intelligible fashion and written in standard English?

Reviewer #1: Yes

Reviewer #2: Yes

5. Review Comments to the Author

Reviewer #1: First impression :

This is an original research on an interesting topic of which we need more data especially from sub-Saharan Africa areas

Overall the paper is adequately structured and well written with minor language and expression editing needed

Abstract

The scope and results of the study are not described clearly

1.First of all , it seems the aims of the study are not presented consistently throughout the paper: Abstract -Line23”In this cross-sectional study at Tikur Anbessa Specialized Hospital Oncology Unit in Addis Ababa,Ethiopia, we assessed breast cancer patients´ perception of barriers towards timely treatment initiation as well as the role of their perceived social support”

Whereas scope of study as presented in introduction : Line 61 “In this study we assessed socio-demographic and clinical factors influencing timely treatment initiation of breast cancer patients at a tertiary referral hospital in Ethiopia and aimed to quantify the influence of patients´ perceived barriers and their perceived social support status towards timely treatment initiation.”

2. The only factors with statistical significance found by this study to influence timely treatment initiation (age and disease stage) are not mentioned in the abstract , neither the median treatment initiation interval

3.line 31”Not being able to pay for the service had been a barrier towards timely care

for 66% of all patients” this is not accurate as which is not accurate as 66% of all patients considered healthcare cost as a perveived barrier but this perception was not related with timely initiation of care - the study failed to show that the perceived barriers had a significant impact untimely treatment

Introduction

Introduction is well written

4.WHO calls for a maximum of 90 days between symptom onset and treatment initiation, whereas the author's definition of late treatment use the pathological diagnosis as a starting point. This is acceptable only if there are no major delays from symptom presentation to diagnosis – I am afraid this may be a major confounding factor as late diagnosis and prolonged time from symptom presentation to diagnosis may also be very common in resource-limited settings

Scopes of the study are differently presented than the abstract (comment No 1)

Methods

Study design and setting: cross-sectional study conducted in two months at the oncology unit of Addis Ababa Hospital stated as the only Hospital in the country offering fully comprehensive cancer treatment . Treatment initiation interval was defined as the interim between pathological diagnosis and the beginning of systemic treatment(comment no 4)

5. The primary outcome-line 72- ( not mentioned in abstract or introduction ) was timely systemic treatment initiation ( chemo or hormonal )which was defined as sooner than 90 days after pathological diagnosis. Background data support the 90 day threshold –again comment no 4) .At this point there is no mention of secondary outcomes like barriers to timely treatment initiation and especially the role of social support

6. There is no mention of radiotherapy which is used early according to breast cancer treatment guidelines in all stages of the disease -was this also assessed? is radiotherapy available in your clinic population/country ?

Population sample and data collection

7. Polulation selection : random sampling (possible bias)? Why not consequitive patients attending the clinic fulfilling inclusion criteria ?

8. Questionnaire procedures should be more detailed: were the questionnaires handed over to patients and self-filled ?or was there an interview? And if that is the case , how many people were involved? Were they always the same conducting the interview ?

Choice of the Pechanski “concept of access” is reasonable and useful for this purpose

9. line 88 what were the data from patient charts that were used, they should be noted in this section

Data analysis

10. Line 97 “To compensate for large outliers IQR are were used to describe intervals” but there is no mention of this analysis in the results section

11. Line 100 that patients whose interval was longer than 2 years were excluded on the assumption that untreated patients will probably have died after 2 years post diagnosis. It is not clear to me how a patient attending the Oncology Clinic thus very much alive would be excluded from the study on these grounds. If it was a decision in order to avoid the extreme outliers I can see the point but the explanation given is not solid .This should have been written in the population selection section and not the data analysis.

12.The demographic and medical factors that were assessed should be stated in the data collection

The data analysis and statistic tools used are appropriate and adequate for this study

Ethical clearance or rather approval of Ethics Committee of local University was obtained

Results

Tables are clear, accurate and easy to read

13. Dates of diagnosis or treatment were known for only two-thirds of the clinic patients

14.Line 117 “use of traditional treatment” : what does that mean exactly? is it something that is very common and people tend to rely on that sort of treatment in Ethiopia? if that is the case its use can be a barrier to timely assessment of standard treatment (traditional treatment in table 2 is identical to alternative treatment in table 3?)

15. The patients were separated according to age (> or<45 years old) , I am guessing this was done for convenience in the analysis.However, wouldn’t it be more accurate to separate the patients to premenopausal and post-menopausal as it is usually done in breast cancer cohorts

16. In the study population there is a very high percentage of women below 45 years old .Are there any available data of hormonal receptors status/breast cancer family history/ triple negative breast cancer percentage in this population? This is not the standard age distribution that one would say in in a breast cancer clinic in Europe so it would be a good idea to be given a context about how consistent this is to other countries in the area.(In the discussion )

17. Socio-demographic characteristics in a cohort with breast cancer should include number of children and nulliparity

18. Line 119 –“the median treatment initiation interval in the study cohort was 85 days with 53% of the patients receiving treatment below that threshold” -As the median is very high and extremely close to the cut-off point , one wonders weather we really talking about two separate populations. I think it is of the utmost importance to provide the median interval for those who had timely vs delayed treatment initiation and see if those populations are actually comparable ( eg 80 days vs 95 days are still very close and in that case maybe extreme quartiles should be compared instead). 85 days from pathological diagnosis-(not from symptom onset ) is well beyond WHO threshold (see comment 4)

19. Table 2- what is the unknown treatment given to patients? Again, radiotherapy is not mentioned on this table

perceived barriers

20. Affordability and accessibility what are perceived as most important barriers 66% of all patients considered not being able to pay for the service however there is no information about the service cost as opposed to medium national income (in the discussion)

21. line 130 : long waiting times were perceived by 48% of all women as an important barrier. How long are actually the waiting times? if a patient is trying to book an appointment in the only Oncology Clinic of the country how soon will she get one? Could this be the main factor prolonging intervals to a median of 85 days ?and thus rendering the scope of this study secondary ?

22. No substantial difference in average scores accessing social support between patients -again where are these two populations (timely and delayed treatment) that different? would that analysis be different if it was done in different quartiles?

Discussion

Perceived barriers have not been proven to influence timely initiation of treatment

The interim until treatment initiation is very high in the whole of the study population

23. It is very important to focus on patients with Stage 1 disease as WHO indicates that people with curable cancer should receive treatment within a month 80% of the population stage one for instance or two should receive treatment the aim is to receive treatment within a month-separate analysis?

24. Again no mention of radiotherapy in discussion

There is no screening program in Ethiopia

25. line 162 it is written that that's steep increase in patient volumes at Addis Ababa Hospital over the last years might be one reason why we found nearly half of all patients receiving systemic treatment more than three months after diagnosis. This seems like the major factor delaying initiation of treatment. Is this in fact the most important factor ? as the study failed to show other factors (Comment 21)

The only statistically significant factors related to time of treatment initiation according to this study –age and stage- could very well represent a selection bias from the Oncology clinic side if they have to triage patients due to work overload and few treatment slots.

26. Line 171 “ larger responsibilities at home” for younger patients- however we don't have information about number of children

27. It would be interesting to see in the discussion the issue of primary care and access to breast cancer diagnosis facilities to gain the context. Are the breast cancer patients able to speak to a healthcare provider early on that will explain the seriousness of the situation and explain how timely initiation of treatment is important?

Strengths and limitations-this could be expanded

Conclusion

Good point on need of WHO and stakeholders intervention to reduce delays

OVERALL IMPRESSION AND RECOMMENDATION

This is a retrospective cohort cross sectional study from the only Oncology Clinic of Ethiopia, assessing timely access to breast cancer systemic treatment (<90 days from pathological diagnosis), sociodemographic and medical factors related to timely treatment , as well as self-perceived barriers to timely treatment and self perceived social support status and its relation to treatment timing

The main findings are a high median waiting time of 85 days, with 47% over 90 days, age and disease stage related to timing of treatment initiation , no perceived barriers had any statistical significance to timing of treatment initiation, no difference of social support between group of timely and delayed treatment initiation

The question itself is really important and we do need more data from resource-limited settings-It is very important to see the patients views and perceptions and kudos to the authors who produced this study under such difficult conditions

However , there are a few major issues in my opinion:

• Definition of delay >90 days betwwn pathological diagnosis to treatment initiation –differs from WHO guidance (90 days from symptom to treatment ) This is acceptable only if there are no major delays from symptom presentation to diagnosis – I am afraid this may be a major confounding factor as late diagnosis and prolonged time from symptom presentation to diagnosis may also be very common in resource-limited settings(comment 4)

• Population selection : random and with possible bias (comment 7)

• Median treatment initiation interval in the study cohort was 85 days with 53% of the patients receiving treatment below that threshold” -As the median is very high and extremely close to the cut-off point , one wonders weather we really talking about two separate populations. I think it is of the utmost importance to provide the median interval for those who had timely vs delayed treatment initiation and see if those populations are actually comparable ( eg 80 days vs 95 days are still very close and in that case maybe extreme quartiles should be compared instead). 85 days from pathological diagnosis-(not from symptom onset ) is well beyond WHO threshold (comment 18)

• Long waiting times were perceived by 48% of all women as an important barrier. How long are actually the waiting times? if a patient is trying to book an appointment in the only Oncology Clinic of the country how soon will she get one? Could this be the main factor prolonging intervals to a median of 85 days ?and thus rendering the scope of this study secondary ? it is written that that's steep increase in patient volumes at Addis Ababa Hospital over the last years might be one reason why we found nearly half of all patients receiving systemic treatment more than three months after diagnosis. This seems like the major factor delaying initiation of treatment. Is this in fact the most important factor ? as the study failed to show other factors -The only statistically significant factors related to time of treatment initiation according to this study –age and stage- could very well represent a selection bias from the Oncology clinic side if they have to triage patients due to work overload and few treatment slots. (Comments 21, 25)

I am very sorry to say that I do not feel this study can be published in its current form . However, we very much need studies like this one from resource limited settings to push the stakeholders and governments to help solve the major problems

Reviewer #2: Breast cancer is considered to be one of the best known neoplasms While its incidence

worldwide does not decline, it is also one of the most successfully treated. Yet, its survival

rates vary from country to country, being significantly much higher in developed than in

developing countries.

Timely detection and timely treatment initiation are commonly recognised as significant

determinants of successful treatment. Literature points to a correlation between socio-

demographic, cost and accessibility conditions as factors responsible for delayed detection

and treatment commencement. The reviewed article provides further evidence to substantiate

this position.

In the article entitled 'Perceived barriers to timely treatment initiation and social support status

among women with breast cancer in Ethiopia', Bethel Teshome and her co-workers, present

the results of their inquiry into the causes of the relatively poor breast cancer treatment

outcomes in Ethiopia. Their retrospective, randomised, structured questionnaire-based study

of a representative breast cancer cohort examines patients' perceived barriers to timely

treatment initiation, basing on the 'Concept of Access' model developed by Pechansky et al.

which specifies five dimensions of relationship between patients and the healthcare system,

namely, availability, accessibility, accommodation, affordability and acceptability. While

recognising the relevant importance of the first four of them as patients' perceived barriers,

the authors focus, in particular, on the analysis of the fifth of them, acceptability, that is,

patients' perceived social support status, finding it a major factor contributing to delay in

timely treatment initiation.

By comparing the relevant importance of barriers to timely detection and timely treatment

initiation of breast cancer in Ethiopia, a developing country with an ambitious national cancer

control plan, the article reveals dilemmas faced by healthcare systems in low- and middle-

income countries and indicates areas where expertise, counselling and funds are most needed

to reduce the gap between cancer patients' care and survival rates between developed and

developing countries.

Notwithstanding the limitations of the study resulting from its retrospective and one-hospital

based perceptive, the study deserves attention also or even in particular because of its

emphasis on the social environment and cultural nature of patients' perceived barriers to

timely treatment initiation and hence success of breast cancer treatment.

6. PLOS authors have the option to publish the peer review history of their article (what does this mean?). If published, this will include your full peer review and any attached files.

Reviewer #1: No

Reviewer #2: No

---

## [Author Response · Author response to Decision Letter 0]

24 Aug 2021

Dear Miss Kowalska,

thank you for considering the manuscript “Perceived barriers to timely treatment initiation and social support status among women with breast cancer in Ethiopia” for publication at PLOS One. We have revised the manuscript to meet the journal´s style requirements.

In addition, you will find the study´s questionnaire in Amharic and English uploaded as supplement 1. We added the requested information on the questionnaire development in the method section of the paper (see lines 87 – 92).

Please excuse the wrong information on the availability of our data: The data is available without restrictions and is also attached as a supporting file. 

We´d like to thank the reviewers for their valuable feedback. Please find our responses to their comments on the following pages. Please note that all line references in this letter refer to the clean version of the manuscript. 

Thank you for the re-consideration of this manuscript.

Sincerley,

Dr. Eva Johanna Kantelhardt, MD

Institute for Medical Epidemiology, Biometrics, and Informatics

Martin-Luther-University Halle-Wittenberg

 

RESPONSES TO REVIEWER 1: 

FIRST IMPRESSION:

This is an original research on an interesting topic of which we need more data especially from sub-Saharan Africa areas.

Overall, the paper is adequately structured and well written with minor language and expression editing needed.

Answer: Thank you for acknowledging the importance of the paper – we have performed the required language and expression editing. 

ABSTRACT

Comment 1: The scope and results of the study are not described clearly. 

First of all, it seems the aims of the study are not presented consistently throughout the paper: Abstract, Line 23: “In this cross-sectional study at Tikur Anbessa Specialized Hospital Oncology Unit in Addis Ababa, Ethiopia, we assessed breast cancer patients´ perception of barriers towards timely treatment initiation as well as the role of their perceived social support”.

Whereas scope of study as presented in introduction, Line 61: “In this study we assessed socio-demographic and clinical factors influencing timely treatment initiation of breast cancer patients at a tertiary referral hospital in Ethiopia and aimed to quantify the influence of patients´ perceived barriers and their perceived social support status towards timely treatment initiation.”

Answer: Thank you for this true remarque – the scope was indeed not clearly described. We have now aligned the description of the scope of the study in abstract and introduction (see lines 23 - 26 and lines 62 - 64). 

Comment 2: The only factors with statistical significance found by this study to influence timely treatment initiation (age and disease stage) are not mentioned in the abstract, neither the median treatment initiation interval.

Answer: We agree, this information should be mentioned in the abstract. We added our findings on the role of age and stage, as well as the median treatment initiation interval to the abstract (see lines 32 - 34).

Comment 3: “Not being able to pay for the service had been a barrier towards timely care for 66% of all patients” (Line 31) - this is not accurate as which is not accurate as 66% of all patients considered healthcare cost as a perceived barrier but this perception was not related with timely initiation of care - the study failed to show that the perceived barriers had a significant impact untimely treatment.

Answer: Yes, the applied wording was not accurate. 66 % of all patients perceived not being able to pay for the service as a barrier, however we could not show an influence of this perception on timely treatment initiation. We phrased this observation more accurately (see lines 34 - 35). 

INTRODUCTION

Introduction is well written.

Comment 4: WHO calls for a maximum of 90 days between symptom onset and treatment initiation, whereas the author's definition of late treatment use the pathological diagnosis as a starting point. This is acceptable only if there are no major delays from symptom presentation to diagnosis – I am afraid this may be a major confounding factor as late diagnosis and prolonged time from symptom presentation to diagnosis may also be very common in resource-limited settings

Answer: In the absence of official guidelines, we chose the 90 days cut-off for timely treatment initiation as it is the informal standard of care at Tikur Anbessa Hospital and has been previously used in comparable studies. We explain this more clearly in the updated manuscript (see lines 74 - 75). As correctly mentioned by reviewer 1, our interval definition is not comparable to the WHO´s recommendation of 90 days between first symptom appraisal and treatment initiation. This recommendation was only mentioned in the paper to underline the ultimate goal, well knowing, that it is still far away from the current reality of breast cancer patients in Ethiopia. We have removed the reference to the WHO´s recommendation from the introduction to avoid confusion. 

METHODS

Study design and setting: cross-sectional study conducted in two months at the oncology unit of Addis Ababa Hospital stated as the only hospital in the country offering fully comprehensive cancer treatment. Treatment initiation interval was defined as the interim between pathological diagnosis and the beginning of systemic treatment (comment no 4).

Comment 5: The primary outcome-line 72- (not mentioned in abstract or introduction) was timely systemic treatment initiation (chemo or hormonal) which was defined as sooner than 90 days after pathological diagnosis. Background data support the 90-day threshold –again comment no 4). At this point there is no mention of secondary outcomes like barriers to timely treatment initiation and especially the role of social support.

Answer: The primary outcome was added to abstract (see linea 23 - 26) and introduction (see line 62 - 64). Secondary outcomes such as barriers to timely treatment initiation and the role of social support are mentioned in lines 91 - 92 and 98 - 102 of the methods section. 

Comment 6: There is no mention of radiotherapy which is used early according to breast cancer treatment guidelines in all stages of the disease -was this also assessed? is radiotherapy available in your clinic population/country?

Answer: Thank you for this important remarque. In line 76 -78 we added the explanation, why patients receiving radiotherapy were not included into the study: 

“Patients receiving radiotherapy were not included into the study, as, with only one radiotherapy machine in the country, the number of breast cancer patients being treated with radiation within the study period was judged to be too small to receive reliable results.”

Population sample and data collection

Comment 7: Population selection: random sampling (possible bias)? Why not consecutive patients attending the clinic fulfilling inclusion criteria?

Answer: We are aware, that random sampling comes with possible biases. However, for the extend of this study we judged it to be the most reliable form of sampling. In lines 85 – 87 we elaborate the sampling method. 

Comment 8: Questionnaire procedures should be more detailed: were the questionnaires handed over to patients and self-filled? Or was there an interview? And if that is the case, how many people were involved? Were they always the same conducting the interview?

Answer: This is true – we originally did not give enough details on the data collection process. Please find a more detailed description in lines 87 - 92, as well as the Amharic and the English version of the questionnaire in the supplements. 

Choice of the Pechanski “concept of access” is reasonable and useful for this purpose

Comment 9: Line 88 what were the data from patient charts that were used, they should be noted in this section.

Answer: We added details on data cross-checking in lines 96 - 97.

Data analysis

Comment 10: Line 97 “To compensate for large outliers IQR were used to describe intervals” but there is no mention of this analysis in the results section

Answer: IQRs and the total ranges of treatment initiation intervals can be found in results, table 2. 

Comment 11: Line 100 that patients whose interval was longer than 2 years were excluded on the assumption that untreated patients will probably have died after 2 years post diagnosis. It is not clear to me how a patient attending the Oncology Clinic thus very much alive would be excluded from the study on these grounds. If it was a decision in order to avoid the extreme outliers, I can see the point but the explanation given is not solid. This should have been written in the population selection section and not the data analysis.

Answer: Thank you for this valuable comment. We excluded patients with intervals longer than 2 years to avoid extreme outliers – however we cannot draw a conclusion about their health status. We aligned this explanation in lines 82 - 84. 

Comment 12: The demographic and medical factors that were assessed should be stated in the data collection. 

Answer: Please find all demographic and medical factors assessed in the questionnaires serving as supplement to this paper (see S1 Questionnaire). 

The data analysis and statistic tools used are appropriate and adequate for this study. Ethical clearance or rather approval of Ethics Committee of local University was obtained. 

RESULTS

Tables are clear, accurate and easy to read

Comment 13: Dates of diagnosis or treatment were known for only two-thirds of the clinic patients.

Answer: This leads to only 196 patients being included into the analysis. We discuss this small sample size within the limitations of this study (see line 214 - 216). 

Comment 14: Line 117 “use of traditional treatment” : what does that mean exactly? is it something that is very common and people tend to rely on that sort of treatment in Ethiopia? if that is the case its use can be a barrier to timely assessment of standard treatment (traditional treatment in table 2 is identical to alternative treatment in table 3?)

Answer: Thank you for this true comment. We have aligned our phrasing and explain the meaning of alternative treatment upon first appearance in the paper (see table 2). 

Comment 15: The patients were separated according to age (> or<45 years old), I am guessing this was done for convenience in the analysis. However, wouldn’t it be more accurate to separate the patients to premenopausal and post-menopausal as it is usually done in breast cancer cohorts.

Answer: 45 years is commonly judged to be the median age of menopause in this population. Due to missing data on the actual menopausal status, we chose this statistical value. We added this information in the result section, table 3.

Comment 16: In the study population there is a very high percentage of women below 45 years old. Are there any available data of hormonal receptors status/breast cancer family history/ triple negative breast cancer percentage in this population? This is not the standard age distribution that one would say in in a breast cancer clinic in Europe so it would be a good idea to be given a context about how consistent this is to other countries in the area (in the discussion).

Answer: Thank you for this comment. A median age of 40 years in a breast cancer cohort is not uncommon in this population due to the very young population structure. We added this point to the discussion and supported it with data from other countries in Sub-Sahara Africa (see line 155 - 156). 

Comment 17: Socio-demographic characteristics in a cohort with breast cancer should include number of children and nulliparity.

Answer: This is true and we mention this limitation in the discussion (see line 177).

Comment 18: Line 119 –“the median treatment initiation interval in the study cohort was 85 days with 53% of the patients receiving treatment below that threshold” -As the median is very high and extremely close to the cut-off point , one wonders weather we really talking about two separate populations. I think it is of the utmost importance to provide the median interval for those who had timely vs delayed treatment initiation and see if those populations are actually comparable ( eg 80 days vs 95 days are still very close and in that case maybe extreme quartiles should be compared instead). 85 days from pathological diagnosis-(not from symptom onset ) is well beyond WHO threshold (see comment 4).

Answer: This question is very justified. We have added the median treatment initiation intervals of both groups to table 2 (32.5 days in the timely group vs. 158.5 days in the not timely group) and judge this to prove the difference between the two groups (see table 2). 

Comment 19: Table 2- what is the unknown treatment given to patients? Again, radiotherapy is not mentioned on this table.

Answer: For patients listed with unknown treatment in table 2 we had inconsistent data on type of treatment from patients´ charts and the face-to-face interviews. We added this information to table 2. 

Comment 20: Affordability and accessibility what are perceived as most important barriers 66% of all patients considered not being able to pay for the service however there is no information about the service cost as opposed to medium national income (in the discussion).

Answer: We discuss the role of affordability and accessibility of treatment in the discussion of this paper (see line 186 -192). The relation of the actual service cost as opposed to medium national income is very complex and currently assessed by a PhD student of our working group. 

Comment 21: line 130 : long waiting times were perceived by 48% of all women as an important barrier. How long are actually the waiting times? if a patient is trying to book an appointment in the only Oncology Clinic of the country how soon will she get one? Could this be the main factor prolonging intervals to a median of 85 days ?and thus rendering the scope of this study secondary ?

Answer: Thank you for this valuable comment. We admit to not have discussed this factor enough regarding its impact on timely treatment initiation. We have added a section on the role of waiting times in the revised version of the manuscript (see lines 193 - 198). However, our data does not allow us to distinguish between a “patient interval” (delay in treatment due to the patient not being able to make an appointment) and a “health care provider interval” (delay in treatment due to challenges on the health system side). We discuss this limitation in the discussion (see lines 217 - 220). 

Comment 22: No substantial difference in average scores accessing social support between patients -again where are these two populations (timely and delayed treatment) that different? would that analysis be different if it was done in different quartiles?

Answer: Please see our answer to comment 18. 

DISCUSSION

Perceived barriers have not been proven to influence timely initiation of treatment. The interim until treatment initiation is very high in the whole of the study population.

Comment 23: It is very important to focus on patients with Stage 1 disease as WHO indicates that people with curable cancer should receive treatment within a month 80% of the population stage one and two for instance should receive treatment the aim is to receive treatment within a month-separate analysis?

Answer: We agree on the importance of treating patients in early stages. However only 29 patients in our study were diagnosed at stage I – we judge this number too small for any form of reliable sub analysis. 

Comment 24: Again, no mention of radiotherapy in discussion

Answer: Please see our answer on comment 6. 

There is no screening program in Ethiopia

Comment 25: Line 162 it is written that that's steep increase in patient volumes at Addis Ababa Hospital over the last years might be one reason why we found nearly half of all patients receiving systemic treatment more than three months after diagnosis. This seems like the major factor delaying initiation of treatment. Is this in fact the most important factor ? as the study failed to show other factors (Comment 21)

The only statistically significant factors related to time of treatment initiation according to this study –age and stage- could very well represent a selection bias from the Oncology clinic side if they have to triage patients due to work overload and few treatment slots.

Answer: We agree, the finding that young age and early stage are related to delayed treatment initiation might be explained by the limited capacity of the oncology unit and the resulting need to triage. However, at TASH there are no policies or triage criteria in place – further studies need to investigate the findings (see lines 182 - 185). 

Comment 26: Line 171 “ larger responsibilities at home” for younger patients- however we don't have information about number of children.

Answer: This is true. We have underlined our inability to prove this assumption in the discussion (see line 177). 

Comment 27: It would be interesting to see in the discussion the issue of primary care and access to breast cancer diagnosis facilities to gain the context. Are the breast cancer patients able to speak to a healthcare provider early on that will explain the seriousness of the situation and explain how timely initiation of treatment is important?

Answer: The role of primary care is a very important topic. It is currently being assessed by two masters and a PhD candidates within our working group. 

Comment 28: Strengths and limitations-this could be expanded.

Answer: We agree on this and have added numerous points to the limitations section of this study (see lines 214 - 220). 

Conclusion

Good point on need of WHO and stakeholders intervention to reduce delays.

OVERALL IMPRESSION AND RECOMMENDATION:

This is a retrospective cohort cross sectional study from the only Oncology Clinic of Ethiopia, assessing timely access to breast cancer systemic treatment (<90 days from pathological diagnosis), sociodemographic and medical factors related to timely treatment, as well as self-perceived barriers to timely treatment and self-perceived social support status and its relation to treatment timing.

The main findings are a high median waiting time of 85 days, with 47% over 90 days, age and disease stage related to timing of treatment initiation, no perceived barriers had any statistical significance to timing of treatment initiation, no difference of social support between group of timely and delayed treatment initiation.

The question itself is really important and we do need more data from resource-limited settings-It is very important to see the patients views and perceptions and kudos to the authors who produced this study under such difficult conditions

However, there are a few major issues in my opinion:

Issue 1: Definition of timely systemic treatment initiation

Definition of delay >90 days between pathological diagnosis to treatment initiation –differs from WHO guidance (90 days from symptom to treatment) This is acceptable only if there are no major delays from symptom presentation to diagnosis – I am afraid this may be a major confounding factor as late diagnosis and prolonged time from symptom presentation to diagnosis may also be very common in resource-limited settings(comment 4)

Answer: In the absence of official guidelines, we chose the 90 days cut-off for timely treatment initiation as it is the informal standard of care at Tikur Anbessa Hospital and has been previously used in comparable studies. We explain this more clearly in the updated manuscript (see line 74 - 75). As correctly mentioned by reviewer 1, our interval definition is not comparable to the WHO´s recommendation of 90 days between first symptom appraisal and treatment initiation. This recommendation was only mentioned in the paper to underline the ultimate goal, well knowing, that it is still far away from the current reality of breast cancer patients in Ethiopia. We have removed the reference to the WHO´s recommendation from the introduction to avoid confusion.

Issue 2: Sampling method

Population selection: random and with possible bias (comment 7)

Answer: We are aware, that our sample comes with possible biases. In lines 85 - 87 we elaborate the sampling method. 

We also discuss the sampling method, exclusion of patients due to missing data and the possibly resulting biases within the limitations of this paper. 

Issue 3: Difference between timely vs. not timely 

Median treatment initiation interval in the study cohort was 85 days with 53% of the patients receiving treatment below that threshold” -As the median is very high and extremely close to the cut-off point , one wonders whether we are really talking about two separate populations. I think it is of the utmost importance to provide the median interval for those who had timely vs delayed treatment initiation and see if those populations are actually comparable ( eg 80 days vs 95 days are still very close and in that case maybe extreme quartiles should be compared instead). 85 days from pathological diagnosis-(not from symptom onset ) is well beyond WHO threshold (comment 18)

Answer: This question is very justified. We have added the median treatment initiation intervals of both groups to table 2 (32.5 days in the timely group vs. 158.5 days in the not timely group) and judge this to prove the considerable difference between the two groups (see table 2). 

Issue 4: Role of waiting times

Long waiting times were perceived by 48% of all women as an important barrier. How long are actually the waiting times? if a patient is trying to book an appointment in the only Oncology Clinic of the country how soon will she get one? Could this be the main factor prolonging intervals to a median of 85 days? and thus rendering the scope of this study secondary? it is written that that's steep increase in patient volumes at Addis Ababa Hospital over the last years might be one reason why we found nearly half of all patients receiving systemic treatment more than three months after diagnosis. This seems like the major factor delaying initiation of treatment. Is this in fact the most important factor? as the study failed to show other factors.

Answer: Thank you for this valuable comment. We admit to not have discussed this factor enough regarding its impact on timely treatment initiation. We have added a section on the role of waiting times in the revised version of the manuscript (see lines 193 - 198). However, our data does not allow us to distinguish between a “patient interval” (delay in treatment due to the patient not being able to make an appointment) and a “health care provider interval” (delay in treatment due to waiting times). We discuss this limitation in the discussion (see lines 217 - 220). 

Issue 5: Role of age and stage

The only statistically significant factors related to time of treatment initiation according to this study –age and stage- could very well represent a selection bias from the Oncology clinic side if they have to triage patients due to work overload and few treatment slots. (Comments 21, 25)

Answer: We agree, the finding that young age and early stage are related to delayed treatment initiation might be explained by the limited capacity of the oncology unit. According to the local guidelines, there is no triage system in place. Therefore, other reasons for these findings have to be explored (see lines 182 - 185). 

I am very sorry to say that I do not feel this study can be published in its current form. However, we very much need studies like this one from resource limited settings to push the stakeholders and governments to help solve the major problems.

Answer: Thank you for this pointed and appropriate feedback. We hoped to have been able to add information wherever needed and answer to all unclear points within the paper as well as in this letter. 

RESPONSES TO REVIEWER 2: 

Breast cancer is considered to be one of the best-known neoplasms. While its incidence worldwide does not decline, it is also one of the most successfully treated. Yet, its survival rates vary from country to country, being significantly much higher in developed than in developing countries.

Timely detection and timely treatment initiation are commonly recognised as significant determinants of successful treatment. Literature points to a correlation between socio-demographic, cost and accessibility conditions as factors responsible for delayed detection and treatment commencement. The reviewed article provides further evidence to substantiate this position.

In the article entitled 'Perceived barriers to timely treatment initiation and social support status among women with breast cancer in Ethiopia', Bethel Teshome and her co-workers, present the results of their inquiry into the causes of the relatively poor breast cancer treatment outcomes in Ethiopia. Their retrospective, randomised, structured questionnaire-based study of a representative breast cancer cohort examines patients' perceived barriers to timely treatment initiation, basing on the 'Concept of Access' model developed by Pechansky et al. which specifies five dimensions of relationship between patients and the healthcare system, namely, availability, accessibility, accommodation, affordability and acceptability. While recognising the relevant importance of the first four of them as patients' perceived barriers, the authors focus, in particular, on the analysis of the fifth of them, acceptability, that is,

patients' perceived social support status, finding it a major factor contributing to delay in timely treatment initiation.

By comparing the relevant importance of barriers to timely detection and timely treatment initiation of breast cancer in Ethiopia, a developing country with an ambitious national cancer control plan, the article reveals dilemmas faced by healthcare systems in low- and middle-income countries and indicates areas where expertise, counselling and funds are most needed to reduce the gap between cancer patients' care and survival rates between developed and developing countries.

Notwithstanding the limitations of the study resulting from its retrospective and one-hospital based perceptive, the study deserves attention also or even in particular because of its emphasis on the social environment and cultural nature of patients' perceived barriers to timely treatment initiation and hence success of breast cancer treatment.

Answer: Thank you for this valuable feedback and the recognition of the importance of this paper. We address the study´s limitations due to its retrospective and hospital-based perspective in the discussion section of this paper. Nevertheless, we agree with reviewer 2 on the relevance of this paper for further scientific and political interventions addressing timely treatment for women with breast cancer in Ethiopia.

---

## [Editor Report · Decision Letter 1]

25 Aug 2021

Perceived barriers to timely treatment initiation and social support status among women with breast cancer in Ethiopia

PONE-D-21-16175R1

Dear Dr. Kantelhardt,

We’re pleased to inform you that your manuscript has been judged scientifically suitable for publication and will be formally accepted for publication once it meets all outstanding technical requirements.

Kind regards,

Justyna Dominika Kowalska

Academic Editor

PLOS ONE

---

## [Editor Report · Acceptance letter]

3 Sep 2021

PONE-D-21-16175R1 

Perceived barriers to timely treatment initiation and social support status among women with breast cancer in Ethiopia 

Dear Dr. Kantelhardt:

I'm pleased to inform you that your manuscript has been deemed suitable for publication in PLOS ONE. Congratulations! Your manuscript is now with our production department. 

Kind regards, 

on behalf of

Dr. Justyna Dominika Kowalska 

Academic Editor

PLOS ONE